# Modeling the efficacy of CRISPR gene drive for snail immunity on schistosomiasis control

**Richard E. Grewelle** [1,2]*, **Javier Perez-Saez**[3], **Josh Tycko**[4], **Erica K. O. Namigai**[5], **Chloe G. Rickards**[6], **Giulio A. De Leo** [1,2,7]*

**1** Department of Biology, Stanford University, Stanford, California, United States of America, **2** Hopkins Marine Station, Stanford University, Pacific Grove, California, United States of America, **3** Department of Epidemiology, Johns Hopkins Bloomberg School of Public Health, Baltimore, Maryland, United States of America, **4** Department of Genetics, Stanford University, Stanford, California, United States of America, **5** Department of Zoology, University of Oxford, Oxford, United Kingdom, **6** Department of Biology, University of California Santa Cruz, Santa Cruz, California, United States of America, **7** Woods Institute for the Environment, Stanford University, Stanford, California, United States of America

\* regrew@stanford.edu (REG); deleo@stanford.edu (GADL)

**Data Availability Statement:** All relevant data are within the manuscript and its Supporting information files. Software and supporting code for

## Abstract

CRISPR gene drives could revolutionize the control of infectious diseases by accelerating the spread of engineered traits that limit parasite transmission in wild populations. Gene drive technology in mollusks has received little attention despite the role of freshwater snails as hosts of parasitic flukes causing 200 million annual cases of schistosomiasis. A successful drive in snails must overcome self-fertilization, a common feature of host snails which could prevents a drive's spread. Here we developed a novel population genetic model accounting for snails' mixed mating and population dynamics, susceptibility to parasite infection regulated by multiple alleles, fitness differences between genotypes, and a range of drive characteristics. We integrated this model with an epidemiological model of schistosomiasis transmission to show that a snail population modification drive targeting immunity to infection can be hindered by a variety of biological and ecological factors; yet under a range of conditions, disease reduction achieved by chemotherapy treatment of the human population can be maintained with a drive. Alone a drive modifying snail immunity could achieve significant disease reduction in humans several years after release. These results indicate that gene drives, in coordination with existing public health measures, may become a useful tool to reduce schistosomiasis burden in selected transmission settings with effective CRISPR construct design and evaluation of the genetic and ecological landscape.

## Author summary

CRISPR gene drives can propagate engineered traits in vectors, like mosquitoes, to curb transmission of infectious diseases. Here we explore whether gene drive technology can also be used in molluscs to control schistosomiasis, a debilitating neglected tropical disease requiring freshwater snails as intermediate hosts. Unlike mosquitoes, these snail species can reproduce by self-fertilization, which could disable gene drive inheritance. Despite this limitation, our mathematical model identifies conditions in which drive

simulations is available at https://github.com/grewelle/ModelGeneDriveSchisto.

**Funding:** REG was funded by the Stanford Graduate Fellowship and ARCS Fellowship; https://ed.stanford.edu/academics/doctoral-handbook/financialsupport/stanfordfellowships, https://vpge.stanford.edu/fellowships-funding/achievement-rewardscollege/details; and the Stanford-EPFL exchange fellowship, https://neuroscience.stanford.edu/research/programs/epfl-stanford-exchange-program. GADL and REG were partially supported by the National Science Foundation's grants DEB-2011179 and ICER-2024383. The funders had no role in study design, data collection and analysis, decision to publish, or preparation of the manuscript.

**Competing interests:** I have read the journal's policy and the authors of this manuscript have the following competing interests: JT and EKON were seed funded by the Merck Innovation Cup 2016 for research on schistosomiasis, and previously employed as external consultants to the Global Health Institute of Merck (KGaA) which produces treatments for schistosomiasis.

immunity in snails can spread and, when complemented with mass Praziquantel treatment, achieves sustained disease reduction. Modeling that integrates genetic designs with ecological conditions and public health interventions is critical to safely leverage a powerful technology like gene drive.

## Introduction

Gene drive technology is rapidly expanding since the discovery of CRISPR-Cas9 [1–3]. Its potential uses include controling diseases, invasive species, and pests by spreading targeted genes through a population faster than traditional Mendelian inheritance allows [4]. For example, there are currently large efforts to harness genetic technology targeting mosquito species that are vectors of malaria and other vector-borne diseases [5–9]. Similar efforts could be on the horizon for schistosomiasis, a debilitating disease of poverty caused by blood flukes of the genus Schistosoma [10].

The battle to eliminate schistosomiasis has been waged for more than a century, and despite local successes, the disease remains widespread [11]. Globally over 200 million individuals are actively infected. With 800 million people at risk of infection, schistosomiasis is second only to malaria in the breadth of its health and economic impact as an infectious tropical disease [12, 13]. The disease manifests as a complex suite of symptoms stemming primarily from the inflammatory processes the body mounts in response to the schistosome eggs that embed in tissue [14]. Abdominal pain, release of blood in urine or stool, fever, enlargement of liver or spleen, and accumulation of fluid in the peritoneal cavity are acute symptoms, while fibrosis and lesions of vital organs, infertility, and bladder and colorectal cancer are lasting consequences of infection [15, 16].

Transmission of schistosomes to intermediate, obligate snail hosts occurs when eggs shed in urine or feces from infected people contact freshwater and emerge as free-swimming miracidia. Once established within the snail, the parasite reproduces asexually and cercariae are released 3–5 weeks after the onset of infection. In this stage, the parasites castrate the freshwater snails, severely reducing reproduction [17]. Released cercariae can penetrate the skin of humans in contact with infested water bodies and cause infection (Fig 1) [18].

Rapid advancements in genomics for the intermediate snail host species provides a mechanistic understanding of innate, genetically-based snail immunity to schistosome infection [19]. Genes responsible for immunity could be candidates for gene drive mediated spread through snail host populations. Promisingly, selection experiments reveal rapid evolution of immune phenotypes, demonstrating high immunity can be achieved under laboratory conditions within a few snail generations [20–22] (Fig 2). Overall, there is good reason to expect that a population modification CRISPR gene drive designed to provide greater immunity in the snail population could soon be developed. However, whether such a gene drive could provide the sustained reduction in transmission necessary to eliminate schistosomiasis in realistic settings laden with barriers to the spread of a drive remains unknown.

Previous theoretical work using classical population genetic models has explored how fitness, homing efficiency, selfing, resistance allele formation, gene flow, and other forms of population structure influence invasion success and peak frequency of a drive in general contexts [23–27]. Stochastic Moran models or discrete deterministic models with non-overlapping generations do not incorporate population dynamics on which the tempo of evolution is highly dependent. Snail populations are iteroparous, reproducing several times within a lifetime, and exhibit density dependent recruitment. This form of reproduction is not modeled in the

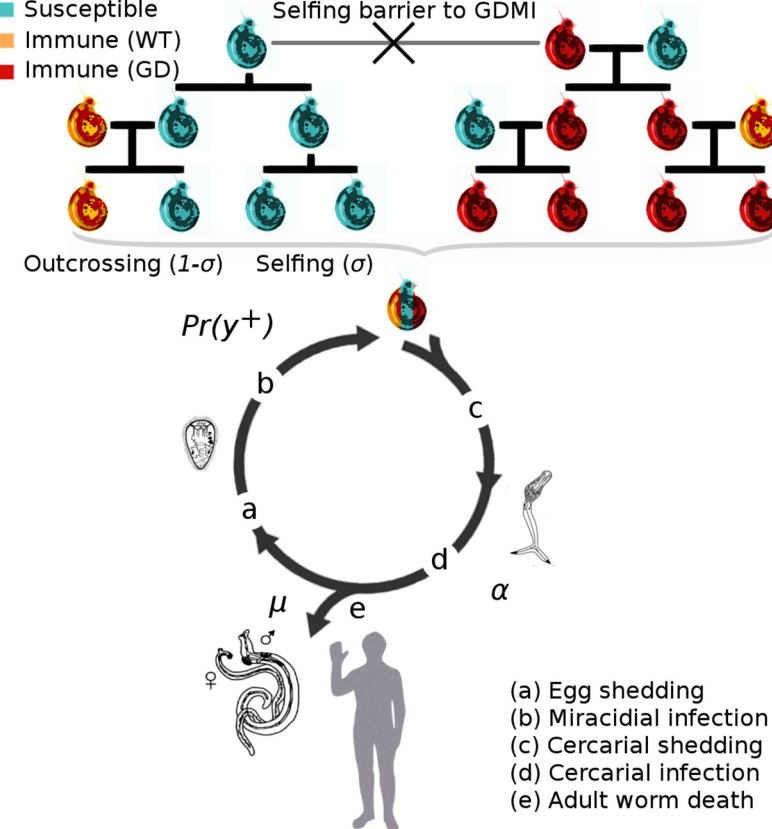

**Fig 1. Conceptual diagram of the integrated epidemiological and population genetic model describing the evolution of immunity to schistosome infection in the snail population.** (a) High worm burden in the human population increases the force of infection on the snail population, which positively selects for immune snail genotypes. (b) Miricidial infection of susceptible snails is density-dependent. (c) Evolution of immunity in the snail population reduces cercarial transmission to humans, thereby regulating parasite densities at an endemic equilibrium. GDMI is inherited more rapidly than natural immunity only when outcrossing occurs. (d) Infection of humans is proportional to cercarial output, and a negative binomial distribution of adult worms in the human population influences mating success and egg production (Table B in S1 Text). Existing literature does not consistently support the role of variable human immunity in epidemiological models in endemic regions, therefore immunity in the human population is assumed constant. (e) Mortality of adult worms occurs via constant natural mortality and MDA treatment. Three snail genotypes are modeled: susceptible to infection, innately immune (wild type), and gene drive mediated immune. Iteroparitive reproduction and mortality of these genotypes is modeled with explicit fecundity and viability components of fitness (see S1 Text).

simplified evolutionary models developed for gene drives to date. Accuracy of gene drive models hinges on realistic assumptions of the target population.

The success of gene drive mediated immunity (GDMI) in natural snail populations is determined by features intrinsic to the design of the drive construct and its deployment—homing efficiency, fitness cost of the payload, evolution of resistance to the drive, and number of releases—and by extrinsic properties of the environment in which GDMI is deployed, such as the size of the focal snail population, transmission rates, and gene flow and standing genetic variation for immunity in snail populations. Importantly, all snail species that serve as intermediate hosts to schistosomes, except for *Oncomelania spp.*, are simultaneous hermaphrodites capable of self-fertilization (selfing). In contrast to mosquito and fruit fly models for which gene drives have been designed, selfing snail species may be incapable of propagating a drive

construct. Gene drive relies on an encoded endonuclease, such as Cas9, which introduces a double strand break in the homologous chromosome that is repaired using the gene drive allele as a template, thereby copying the gene drive allele to the homologous chromosome [4, 5]. Sexual reproduction (outcrossing) is necessary for gene drive to spread a target allele in a population through pairing and reassortment of gene drive and wild type alleles, facilitating gene conversion. Because the propensity to self-fertilize varies by species and environmental conditions [28], it is imperative to understand how selfing interacts with the variety of intrinsic and extrinsic factors that may influence the establishment of GDMI in natural snail populations.

The impact of GDMI is determined by public health outcomes and not by establishment alone. Local success in schistosomiasis reduction can be achieved through sustained non-phar-maceutical intervention, but such approaches are often resource intensive (e.g. sanitation) or cause collateral damage to the environment (e.g. molluscicides) [29, 30]. Praziquantel (PZQ) emerged in the 1980s as the drug of choice for mass drug administration (MDA) [31, 32], and while cheap and effective in removing mature parasites from infected people and temporarily reducing morbidity, PZQ does not prevent reinfection, and extensive MDA campaigns have been unable to locally eliminate the disease in high transmission regions [33, 34]. For this rea-son, in recent years there has been a renewed interest in complementing MDA with environ-mental interventions aimed at targeting the environmental reservoirs of the disease [35–38]. GDMI has the potential to augment environmental interventions as a means toward cost-effective and sustainable schistosomiasis elimination, especially when paired with existing anthelmintic treatment of humans.

We investigate the role of selfing and its interaction with other factors influencing GDMI establishment to infer the challenges and opportunities for GDMI in a natural context. We hypothesised that a high selfing rate would incapacitate a gene drive, but a lower selfing rate could be compatible with a drive in certain conditions. To test these ideas we developed a bio-logically realistic mathematical model incorporating both genetic and environmental factors. This model is integrated in an epidemiological framework to evaluate the reduction of disease burden in humans with and without coincident MDA treatment. This study can be used as an informative first step for scientists, stakeholders, and policy makers looking to address the large human health crisis of schistosomiasis in conjunction with the principles for responsible use of gene drives proposed by the National Academies of Science, Engineering, and Medicine (NASEM) [39].

## Methods

### Model summary

We present a model of mixed mating strategy and explore a range of observed selfing rates to understand how reproduction strategy influences success of gene drive technology in a natu-ral population. The gene drive model developed embeds a non-stationary Markov process that accounts for natural inheritance patterns as well as gene drive inheritance and fitness dif-ferences among genotypes. In contrast to previous gene drive models, which consider a wild-type allele and gene drive allele, we consider an expanded model with two wildtype alleles: susceptible ($A$) and immune ($B$). A third allele, $B_g$, represents engineered immunity to infec-tion in the form of a gene drive construct. The set of six genotypes formed by these three alleles is $\Omega = \{AA, AB, BB, AB_g, BB_g, B_gB_g\}$. Let $P_i$ be the frequency of each genotype where $i \in \Omega$. Let $\boldsymbol{P_t}$ be the row vector composed of genotype frequencies at time (in generations) $t$. We describe the mixed mating system of genetic inheritance with two transition matrices, $\boldsymbol{S}$ (self-fertilization) and $\boldsymbol{T}$ (out-crossing), to describe the transitional probabilities from generation $t$

to $t + 1$.

$$\mathbf{S} = \begin{pmatrix} 1 & 0 & 0 & 0 & 0 & 0 \\ \frac{1}{4} & \frac{1}{2} & \frac{1}{4} & 0 & 0 & 0 \\ 0 & 0 & 1 & 0 & 0 & 0 \\ \frac{1}{4} & 0 & 0 & \frac{1-H}{2} & 0 & \frac{H}{2}+\frac{1}{4} \\ 0 & 0 & \frac{1}{4} & 0 & \frac{1-H}{2} & \frac{H}{2}+\frac{1}{4} \\ 0 & 0 & 0 & 0 & 0 & 1 \end{pmatrix} \tag{1}$$

$$\mathbf{T} = \begin{pmatrix} P_A & P_B & 0 & (1-H)P_{B_g} & 0 & HP_{B_g} \\ \frac{P_A}{2} & \frac{P_A+P_B}{2} & \frac{P_B}{2} & \frac{(1-H)P_{B_g}}{2} & \frac{(1-H)P_{B_g}}{2} & HP_{B_g} \\ 0 & P_A & P_B & 0 & (1-H)P_{B_g} & HP_{B_g} \\ \frac{P_A}{2} & \frac{P_B}{2} & 0 & \frac{(1-H)(P_{B_g}+P_A)}{2} & \frac{(1-H)P_B}{2} & \frac{H+P_{B_g}}{2} \\ 0 & \frac{P_A}{2} & \frac{P_B}{2} & \frac{(1-H)P_A}{2} & \frac{(1-H)(P_B+P_{B_g})}{2} & \frac{H+P_{B_g}}{2} \\ 0 & 0 & 0 & (1-H)P_A & (1-H)P_B & P_{B_g}(1-H)+H \end{pmatrix} \tag{2}$$

Homing efficiency $H$ takes values between 0 (Mendelian inheritance) and 1 (complete fidelity of gene drive mechanism). A matrix $\mathbf{Q}$ can be formed to represent the mixed mating system with self-fertilization rate and cost of inbreeding given by $\sigma, \xi \in [0, 1]$, respectively.

$$\mathbf{Q} = \sigma(1 - \xi)\mathbf{S} + (1 - \sigma)\mathbf{T} \tag{3}$$

In the absence of population dynamics and fitness differences between genotypes, Eq 4 would suffice to describe genotype frequency changes over time.

$$\mathbf{P}_{t+1} = \mathbf{P}_t\mathbf{Q} \tag{4}$$

These evolving genotypes represent the converted germline rather than embryonic genotypes; however, our default model simulates complete dominance of immunity and therefore the immunity of drive heterozygotes is identical to that of drive homozygotes.

To accurately represent the evolution of traits in the snail-schistosome system, we relax some simplifying assumptions by incorporating fitness differences in response to viability and fecundity selection. We also introduce overlapping generations with density dependent recruitment in the snail population (Equns 8–52 in S1 Text). A simplified model of death, migration, and recruitment is given below in Eq 5:

$$N_i(t + 1) = N_i(t)[1 - \gamma_i(t) + m_i(t)] + \frac{\lambda_i(t)}{\lambda(t)}[G(t) - N_i(t)[1 - \gamma_i(t) + m_i(t)]] \tag{5}$$

Each genotype is indexed by $i$, $N$ represents the population size, $\gamma$ is the finite death rate, $m$ is the finite migration rate, $\lambda$ is the finite growth rate, and $G$ is the logistic growth equation. Model simulations recapitulate laboratory results under the same conditions (Fig 2). We derive analytical solutions at long term equilibrium (Equns 53–62 in S1 Text) and analyze the evolution of resistance to the gene drive mechanism (Equns 63–65 in S1 Text). The model is expanded to simulate the effects of 'daisy chain' drive (Fig B and Equns 66–75 in S1 Text), and invasion analysis is performed for key variables, while a stochastic model is used to observe extinction conditions (Fig H and Equns 76–78 in S1 Text).

The genetic model is integrated with an epidemiological model of schistosomiasis through the fraction of immunity in the snail population, $\rho$.

$$\frac{dw}{dt} = \alpha y - \mu w \tag{6}$$

$$\frac{dy}{dt} = \Lambda^*(1 - e^{-\beta w})(1 - y - \rho) - vy \tag{7}$$

$\alpha$ and $\beta$ are transmission rates governing the conversion of snail infection prevalence, $y$, to adult worms, $w$, in humans and vice versa. $\beta$ is a function of the shape of the negative binomial distribution of adult worms in the human population (Table A in S1 Text). Gurarie et al. outlined the transmission from humans to snails is saturating with increasing worm burden [40]. The asymptote is the pre-treatment force of infection, $\Lambda^*$. $\mu$ and $v$ are the death rates of adult worms and infected snails, respectively.

Eqs 6 and 7 are integrated in 3 month intervals, corresponding to the expected generation time of the snail population. The probability of a new infection per susceptible snail in a generation determines the strength of selection for immunity in the genetic model.

$$Pr(y^+) \approx \int_{t=\tau}^{\tau+1} \Lambda^*(1 - e^{-\beta w})dt \tag{8}$$

Parameter values and initial endemic conditions are detailed in Tables A and B in S1 Text. Calculations for equilibrium values and reproduction numbers are made in Equns 72–91 in S1 Text. The epidemiological model was not used to simulate Figs 3 and 4. Instead, the probability of new infections was held constant at the equilibrium value calculated for endemic conditions with the integrated genetic and epidemiological model (Fig J in S1 Text). Fig 5 incorporates both dynamic models to evaluate single and paired treatment. MDA is modeled as instantaneous annual treatment. Percent reduction in worm burden during each treatment was 60% and is the product of coverage and efficacy.

Python code for simulations is available at www.github.com/grewelle/ModelGene DriveSchisto.

## Model validation with empirical data

Tennessen et al. 2015 [19] performed selection experiments using two infection conditions: 10 and 30 miracidia per snail. These snails were 10 generations from natural *Biomphalaria glabrata* breeding populations and were kept together to breed during each generation of selection. After challenging each group of snails with miracidia, infected snails were removed from the breeding population. Selection for immunity was evident and genetically based as given by experimental evidence of decline in infection through the 6 generations of challenges. We modify our model to replicate these experimental conditions, first by assuming that removal from the breeding pool only occurs via infection (mortality = 0). We assumed a high

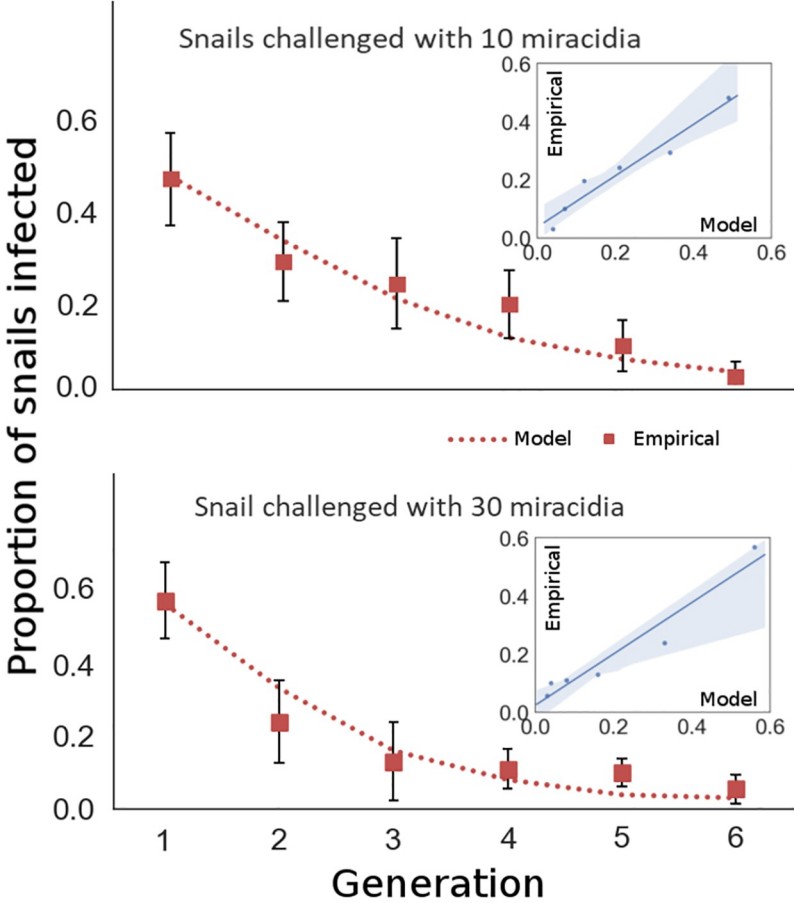

**Fig 2. Comparison of model results to empirical data from published selection experiments by Tennessen et al. 2015.** Two selection experiments were conducted, the first challenging each snail with 10 miracidia (top, $R^2 = 0.944$), and the second challenging each snail with 30 miracidia (bottom, $R^2 = 0.937$). 95% confidence bars are displayed for each empirical measurement. Given initial conditions similar to experimental conditions, both models perform well in recapitulating selection for immunity to infection.

probability of infection of 80% for susceptible snails in the 30 miracidia experimental condition. Because the snails are kept in close proximity, and *B. glabrata* are known to outcross frequently, we assume that outcrossing was the exclusive mode of reproduction (i.e. selfing = 0). Initial allele frequencies were calculated on the basis of the frequency of observed infections (approx. 57%) in the 30 miracidia experimental condition for a probability of infection of 80% for susceptible snails. GDMI was absent and set to a frequency of 0. Otherwise, parameters were unaltered from simulation conditions presented throughout the paper. Initial allele and genotype frequencies were assumed the same between the two experimental treatments, and the probability of infection of susceptible snails was calculated given the frequency of observed infections in the first challenge (48%). The probability of infection for the 10 miracidia treatment is 70%. The curves produced by the model in Fig 2 of expected infection frequencies given these two calculated probabilities (70% and 80%) reflect the observed data well despite some assumptions (e.g. no self-fertilization) and experimental variability. We consider this fit qualitatively similar because some unknown experimental conditions are assumed, and therefore represent one of the possible model outcomes. However, empirical evidence suggesting that immunity is a dominant trait and that it is regulated by a gene complex, which is tightly

linked, corroborates our use of a one locus, complete dominance ($h = 1$) model. Model parameters used in the simulations are able to generate similar evolutionary dynamics to experimentally achieved evolution, and therefore, their values are further supported by our results in addition to support from literature. The values of some genetic model parameters may differ for other snail species or genes conferring immunity than those so far studied for *B. glabrata*.

## Results

We developed a population genetic model that accounts separately for fecundity and viability components of fitness as well as for density dependent dynamics of the snail host population. We expand the wild type—gene drive, 2 allele model to separate the naturally occurring alleles into immune and susceptible types. The resulting six genotypes are formed from three alleles (susceptible, innately immune, gene drive mediated immune) and incorporated into a Markov model modified to include overlapping generations and population dynamics of susceptible and infected snails. Finally, we integrated the population genetic and ecological model with an epidemiological model to describe the dynamics of infection in the human population. Parameter values for the genetic model are derived from literature (Table A in S1 Text) or otherwise explored in sensitivity analyses in the resulting Figs 3 and 4, and under default conditions, simulated evolution recapitulates challenge experiments (Fig 2). We examine the impact that the self-fertilization (selfing) rate of a focus population has on the establishment of gene drive in 10 years.

To represent the two species clusters, we depict results from both ends of the range of observed selfing rates among snail hosts. At high rates ($\sigma = 0.8$), self-fertilization undermines the gene drive and prevents establishment. However, GDMI is able to overcome low rates of self-fertilization ($\sigma = 0.2$) and establish at high frequencies (Fig 3A). Self-fertilization is largely species dependent and may vary with local conditions with higher propensities to self-fertilize observed at low population densities [41]. To simulate this range of conditions, we perform a sensitivity analysis at the 10 year endpoint, so chosen as the likely window in which the efficacy of targeted treatments are evaluated in human populations. Especially for predominantly outcrossing species, reduced offspring viability is associated with self-fertilization [41]. We provide a confidence interval around the endpoint sensitivity analysis based on a range of inbreeding costs to fecundity. Gene drive success in 10 years is highly dependent on low selfing rate, though slower establishment is possible at moderate selfing rates. The inflection point near $\sigma = 0.6$ gives the value over which gene drive success is improbable in a 10 year window (Fig 3B). These results indicate that for species with a lower rate of selfing, including *Oncomelania hupensis*, *Biomphalaria glabrata*, and *Bulinus globosus*, gene drive could establish rapidly in focus populations [42]. Conversely, for species like *Biomphalaria pfeifferi* or *Bulinus truncatus*, which have been observed to self-fertilize at rates higher than 0.6, gene drive will likely be ineffective in increasing immunity in the snail population [43].

Drive success depends on features in the snail-human-schistosome system beyond selfing. Features intrinsic to the design and deployment of the drive like homing efficiency, fitness cost of the payload, resistance evolution, and the number of releases of GDMI individuals are more easily modified than extrinsic factors which are dependent on the ecological and environmental conditions—size of the snail population, force of infection from the human population, gene flow, and standing genetic variation. Yet the success of GDMI may be sensitive to any of these factors. We explore via endpoint sensitivity analyses how variation in these factors alters the frequency of GDMI after 10 years.

Like results from previous modeling and laboratory studies, we find that homing efficiency has a dramatic impact on the outcome of the gene drive release in a focus population. Under

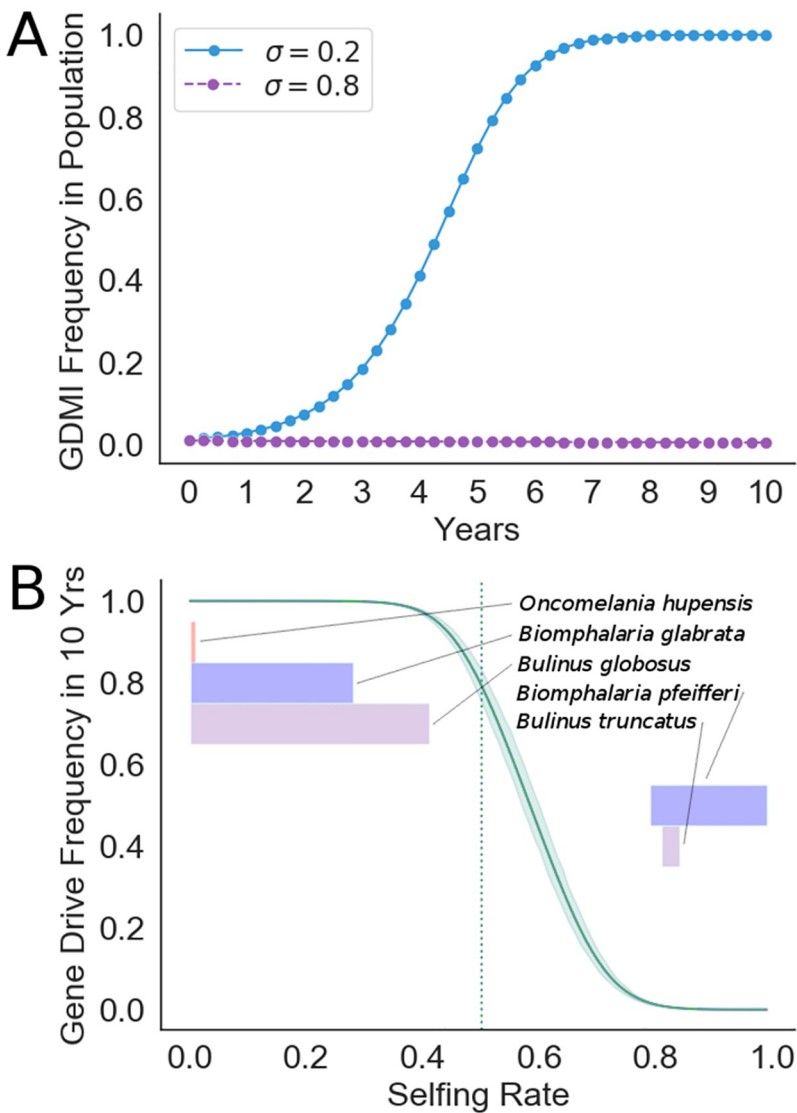

**Fig 3. Self-fertilization rate strongly affects establishment of gene drive in a 10 year window.** (A) Simulation of gene drive invasion under default conditions when self-fertilization rate is low at 0.2 and high at 0.8. (B) Endpoint sensitivity analysis depicting the gene drive frequency in the population after 10 years under variable self-fertilization rates ($\sigma$) from 0 to 1. Bootstrapped 95% confidence intervals are reported on the range of results when the fecundity cost of inbreeding varies on the uniform distribution [0,0.6]. The vertical dotted line designates $\sigma = 0.5$, which is the value used in future simulations to represent an intermediate selfing rate from those observed. Shaded bars colored by genus display ranges (mean ± 1 s.d.) of observed selfing rates for each host snail species for which empirical measures exist [41]. Vertical positioning of the bars is ordered by minimum selfing rate according to the displayed ranges.

the range of selfing scenarios, low homing efficiency leads to minimal gene drive establishment. Laboratory work in mosquitoes and mice shows homing efficiency above 0.4 is achievable and often exceeds 0.9 [5, 44]. In this range, diminishing returns are observed when $H > 0.5$ (Fig 4A). The fitness cost of the genetic payload is not often empirically measured, though for this modeled system, gene drive success is highly sensitive to this parameter: GDMI can establish only when the cost is below 0.4 per gene drive copy in the genome (Fig 4B). Results improve nearly linearly below a fitness cost of 0.3 per copy. In natural and laboratory populations, resistance to the gene drive mechanism can evolve quickly without the presence

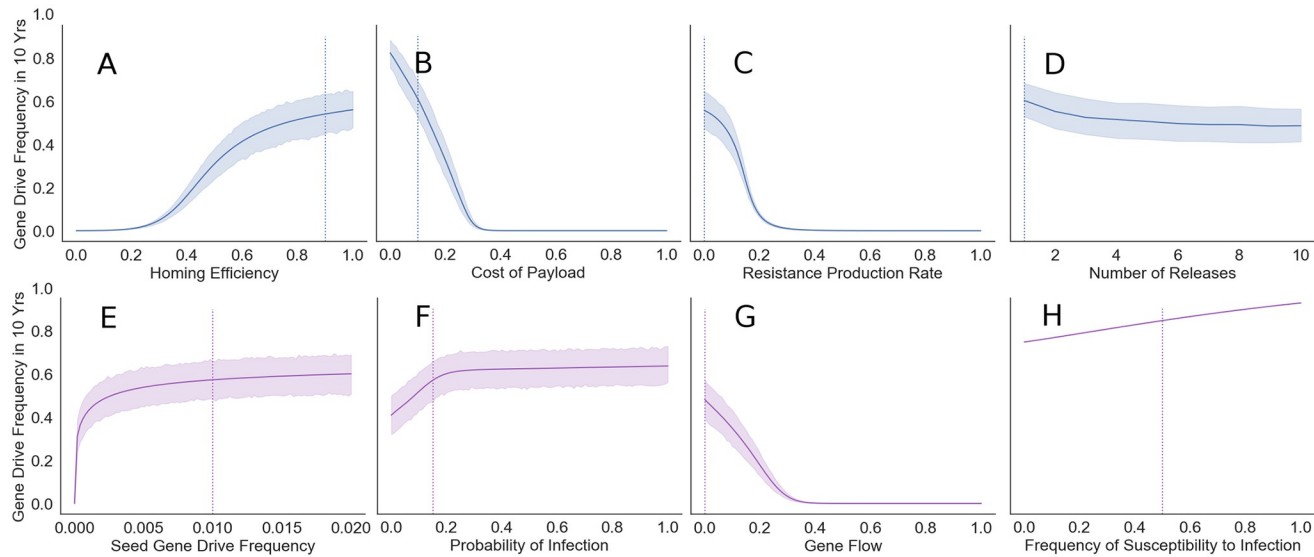

**Fig 4. GDMI efficacy across a wide range of endemic conditions and genetic design.** Bootstrapped 95% confidence intervals of the mean are reported on the range of results when the self-fertilization rate varies on the uniform distribution [0, 1]. Intrinsic factors to the design and release that may be modified are in the top row in blue: (A) Gene drive homing efficiency (homing success per meiotic event), (B) fecundity cost of the genetic payload (relative fitness loss), (C) rate of production of gene drive resistant mutants (resistant mutants per meiotic event), (D) number of annual GDMI partial releases needed to achieve the size of a single maximum release (e.g. 5 = 1/5th release in each of the first 5 years). Extrinsic factors that are not readily modified are in the bottom row in purple: (E) seed frequency (i.e. proportion introduced in a single release at $t = 0$) of the gene drive engineered snails in the population, (F) the probability a susceptible snail is infected in a generation, (G) bidirectional gene flow as the proportion of the focus population that is replaced by an external source each generation, and (H) starting frequency of snails susceptible to infection in focus the population. Because the frequency of the susceptible genotype is a function of selfing rate, bootstrapping is not appropriate for panel H. Vertical dotted lines depict the default parameter values used in the other figures. Note that the solid line represents a mean across uniformly distributed selfing values, $\sigma \in [0, 1]$, rather than the predicted results at $\sigma = 0.5$.

of multiple gRNA or selection against resistance formation [45]. Resistance can evolve more quickly when associated fitness costs of the gene drive phenotype are high. The reported mechanisms of resistance are spontaneous mutation and non-homologous end joining which render the Cas9 cleavage site unrecognisable [46]. We combine these associated mechanisms and display the scenarios for the likely range of summed rates of both processes. In a 10-year time frame, a rate of resistance formation greater than 0.2 per meiotic event makes GDMI establishment infeasible (see also Fig F in S1 Text). With the exception of the deployment strategy, in

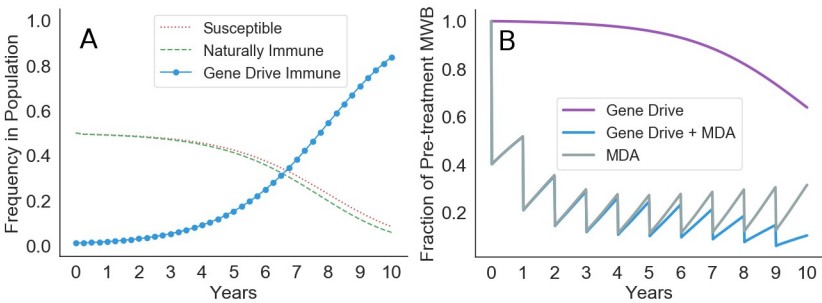

**Fig 5. Combining gene drive with mass drug administration.** (A) 35% reduction in mean worm burden (MWB) is observed in 10 years with gene drive alone. (B) Targeted administration of MDA at 60% annual reduction (efficacy*coverage) in MWB results in more rapid but temporary reduction than the use of gene drive. Sustained reductions are achieved with coincident MDA and gene drive treatment.

which the number of releases does not significantly alter establishment drive success, intrinsic factors to the design of the gene drive construct bear heavily on the outcome of GDMI in 10 years.

We also investigated four extrinsic determinants of GDMI establishment: seed gene drive frequency, force of infection, gene flow, and the standing innate immunity in the snail population. Seed frequency is critical to gene drive spread when only low seed frequencies are possible (Fig 4E). Seeding greater than 1% gives strong diminishing returns. As focus snail populations can vary between hundreds and hundreds of thousands of individuals, this implies that anywhere from one to thousands of snails will need to be raised for a successful introduction, and the size of the focus population will determine the feasibility of release. Similarly, diminishing returns are seen as the probability of infection in a generation increases past 0.2 (Fig 4F). This indicator of endemicity provides the positive selection necessary to propagate the drive in the snail population, as susceptibility to infection is disadvantageous. These results suggest that success is similar for localities experiencing moderate or high burden of disease. Loss of drive alleles from the focus population due to migration inhibits establishment of the drive (Fig 4G). Levels of gene flow greater than 40% (i.e. 40% of focus population alleles are replaced by alleles from a non-evolving background population each generation) bring the drive alleles to undetectable levels assuming immigrants to the focus population lack gene drive immunity. Importantly, GDMI to schistosome infection acts by elevating the level of naturally occurring innate immunity in the snail population. This co-occurring immunity is positively selected under the same conditions as GDMI. Susceptibility is positively selected with weak force of infection due to the fitness costs via reduced egg viability associated with immunity, and immunity is positively selected with moderate to high force of infection due to fitness costs via parasitic castration and reduced lifespan in infected snails [47–49]. High levels of natural immunity will slow the growth of gene drive through direct competition, and therefore, higher susceptibility to infection in a population favors gene drive establishment (Fig 4H). Natural immunity is inherited more slowly, though fecundity for naturally immune snails is assumed higher than GDMI due to added costs of maintaining the genetic payload of the drive.

Although mass drug administration (MDA) is capable of temporary reduction in morbidity, MDA alone is incapable of local elimination at high transmission sites. In these conditions, gene drive offers a potentially promising avenue for coincident MDA and environmental treatment of schistosomiasis. We evaluate the consequences of applying GDMI snails to a community with concurrent annual MDA treatment. We compare the observed reduction in mean worm burden (MWB) between three treatment regimes: gene drive immunity, MDA, and concurrent application of both (Fig 5). Simulations are conducted under the same default conditions evaluated above with the difference that human-to-snail force of infection is a variable that is determined by the number of mated worm pairs in the human population (Table B in S1 Text). The pre-treatment prevalence of infection in humans is assumed to be 80%. The snail-to-human force of infection is a function of the number of infected snails at a water access site, a quantity that diminishes as immunity to infection increases in the snail population.

With gene drive treatment alone a 35% reduction in MWB is observed due to the reduced establishment of new worms in humans and natural mortality of existing adult worms with average lifespan of nearly 5 years [50]. Elimination could be achieved with successful gene drive treatment alone, though the lifespan of adult schistosomes precludes rapid elimination (30 years required for 99% reduction, Fig M in S1 Text). This is true for previously MDA treated and untreated populations (Fig M in S1 Text). With annual reduction of MWB of 60% through targeted MDA, alleviation is possible, but elimination is infeasible due to the

persistence of infected snails in nearby water access sites. Moreover, immunity in snails wanes due to decreased force of infection on the snail population, resulting in an upward trend in MWB from year five onward. Concurrent treatment targeting both snail and human hosts leads to sustained elimination provided resistance formation is low. This is true even when MDA is ended after 10 years because GDMI has reached fixation in the focus snail population (Figs K-M in S1 Text).

## Discussion

Our results demonstrate that successful establishment of immunity within a 10 year evaluation period is possible for species of snails with low to moderate selfing rates. Snails species like *B. pfeifferi* and *B. truncatus*, which are known to self-fertilize at high rates, are likely not desirable targets for GDMI. Many other snail species self-fertilize at lower rates, providing more opportunity for GDMI control of schistosomiasis [28]. Likewise, propensity to self-fertilize can also vary by environmental condition. Panmictic and stable snail populations favor out-crossing, which increases the rate of inheritance of GDMI. This work indicates that the potential for success of GDMI could be evaluated prior to programme implementation through genetic studies quantifying selfing rates (e.g. with F-statistics) in intervention areas. In areas with sympatric snail species with differing selfing rates, quantifying the relative abundance of each species and their respective contributions to schistosomiasis burden will inform the potential for success of GDMI locally. *Schistosoma japonicum* may be treated most effectively with GDMI because *O. hupensis* is not known to self-fertilize, whereas *S. mansoni* and *S. haematobium* are transmitted by snail species with high and low selfing rates, making ecological surveys and feasibility modeling crucial in advance of intervention for these parasite species.

GDMI establishment is sensitive to genetic design and less sensitive to standing genetic variation for immunity. Low payload fitness costs and homing efficiency greater than 50% are essential. Reducing the evolution of resistance to the drive with multiple gRNAs [45] or through other techniques can moderately improve success in a 10 year window and has stronger implications for success after 10 years. Alternative designs incorporating 'daisy chain' inheritance or other drive decay mechanisms can provide safeguards to gene drive release in natural ecosystems, and peak GDMI frequency would be contingent on the strength of this decay, which occurs more quickly with fewer loci in the chain [24]. Selfing requires more loci in a daisy chain to achieve high peak frequencies of GDMI prior to decay, therefore this technology also will perform best for preferentially outcrossing species but will be ineffective for large snail populations (Fig B in S1 Text). Although CRISPR represents the most likely technology for developing GDMI constructs, it may not be the only successful gene editing method; our results cover a wide range of scenarios that also apply to non-CRISPR drives. Other genetic features like dominance, penetrance, and epistatic interactions are significant considerations for choosing appropriate gene targets (Figs C-E in S1 Text). Although optimizing genetic designs is not trivial [51], because modifying snail habitat on a large scale is more challenging, efforts to improve drive construct designs will yield higher returns in successful establishment of GDMI.

Ecological factors that dilute the frequency of gene drive in a focus population (e.g. at a water access site), such as high gene flow due to snail migration or a large snail population size, inhibit timely establishment of gene drive mediated immunity (see also Fig G in S1 Text). These results indicate that success is most easily achieved in isolated water bodies with smaller snail populations. Because selfing is encouraged for isolated snail populations, a trade-off between selfing rate and size of an interbreeding snail population may prove challenging to optimize GDMI introduction. Snail population sizes in many areas fluctuate dramatically by

season [52], therefore introduction of GDMI snails may be best timed when population sizes are at their lowest and have maximum growth potential. Otherwise, GDMI will establish slowly in populations experiencing high seasonality (Fig I in S1 Text). Populations with shorter generation times will achieve greater GDMI frequencies within 10 years (Fig F in S1 Text). Future studies should build on this foundational simulation by considering snail migration and water flow between locations to assess whether GDMI snails would be effective in a wider range of scenarios.

There are some caveats and complexities that we have not addressed here. This model was built on the assumption that the gene drive works, i.e. that the gene drive is effective in producing snails that are immune to schistosome infection. While immune alleles associated with the PTC I and II gene clusters in *B. glabrata* can be rapidly selected in experimental conditions, these alleles have not yet been successfully deployed in a gene drive construct. Unidentified loci could be responsible for immune phenotypes so far examined [53]. Further genetic work is required to discover gene drive targets in other snail host species. Due to the flexibility of the modeling framework, model results can be updated to reflect qualities of constructs in development and the species targeted. In addition, we have not considered potential interactions with other trematode species, as snails can be the intermediate hosts to species other than schistosomes [54]. Interactions between schistosomes, other trematode species, and immunity can shape the fitness landscape in which GDMI operates, and therefore require further investigation to gauge whether the efficacy of this gene drive approach is sensitive to these interactions. This equally applies to interactions that involve schistosome subtypes that may evade a single GDMI design. Field work to identify sympatric schistosome subtypes will be necessary to evaluate local deployment of GDMI. Our analyses primarily explored variation in GDMI success due to genetic parameters, but epidemiological parameters, such as the shape of the distribution of adult worms in the human population or the lifespan of the snails in a local environment, can also influence these results. Although we evaluate the outcomes of GDMI over a range of sensitive conditions, the strength of this model is its flexibility to incorporate emerging data to simulate local conditions.

These results indicate that the use of GDMI together with MDA could contribute to a longer-lasting reduction of worm burden than either GDMI or MDA alone. This emphasises that gene drive is one potential tool among several that are currently available, and optimal use would likely be in conjunction with current control methods. GDMI is much more targeted than molluscicides, as it does not destroy the populations of snails and other aquatic life, and thus may be preferred by many stakeholders. Moving forward, it will be necessary to model how gene drives could interact with the variety of other control methods, to assess the optimal combination of methods and timing that would result in sustained elimination.

Modeling is crucial to understand the feasibility of implementing a new technology like gene drive, particularly in a natural system. Although this technology represents a new frontier for controling disease, pests, and invasive species, the spread of designed genes in a natural setting can carry serious ethical and practical implications [39, 55]. It is therefore prudent to begin any considerations with *in silico* and *in vitro* studies, before proceeding to *in vivo*, with earlier steps informing the next. Further, modeling can ground critical deliberations amongst stakeholders by providing realistic predictions for the effects of a gene drive project and provide a useful ability to rapidly perform new simulations to address questions that stakeholders might have for gene drive developers [56]. This model is an advancement towards a biologically realistic simulation which integrates population genetics, epidemiology, and population dynamics, and can serve as a template for future work in gene drive feasibility analysis.

## Supporting information

**S1 Text. Detailed methods, including analytical derivations of genetic and epidemiological models.** Additional genetic results, including the evolution of resistance to a drive, daisy chain drives of 1–5 loci, and the sensitivity of GDMI to a variety of genetic variables. Additional epidemiological results, including invasion and extinction analyses as well as the sensitivity of GDMI to a variety of epidemiological variables. Fig A. Forward simulations under fixed epidemiological conditions of the spread of GDMI with various resistance production rates per homing event. (A) No resistant alleles are produced. (B) Resistant alleles are produced with 20% of homing events. GDMI achieves only low frequency in the population due to rapid evolution of resistance to the drive mechanism. (C) Resistant alleles are produced with 10% of homing events. GDMI rises slowly, achieving half the frequency in the population compared to conditions where resistance does not evolve. (D) Resistant alleles are produced with 10% of homing events as in panel C. In 20 years it is evident that the frequency of resistant alleles outpaces the homing efficiency benefits in inheritance of GDMI, and GDMI declines after reaching intermediate frequency (eventually to negligible frequency). Fig B. Forward simulations of daisy drive systems for the inheritance of GDMI designed with 1–5 daisy chain loci. Decay of the drive occurs after $n$ generations, therefore more loci produce a longer lasting drive. However, because GDMI spreads slowly in the population compared to a fully outcrossed population, peak frequency of GDMI is low. Nearly 30 daisy chain loci are required to reach peak frequency of 50%, rendering daisy drive infeasible for implementation in this system. Fig C. The relationship between the immune and susceptible alleles described by the dominance coefficient governs the trajectory of evolution for naturally-occurring immunity. Lower dominance of the immune allele leads to slower evolution of immunity, which could change the speed at which GDMI increases in frequency in a population. Fig D. High dominance (top panel, h = 1) representing PTC 1 and low dominance (bottom panel, h = 0.4) representing PTC 2 do not yield measurably different results under default simulation conditions after 10 years. Fig E. The effect of default penetrance ($\iota = 0.8$) compared to higher penetrance ($\iota = 0.9$) in the establishment of GDMI. Higher penetrance produces the blue GDMI and red susceptible lines, while lower penetrance produces the orange GDMI and light blue susceptible lines. Fig F. Simulations of susceptible and GDMI frequencies under variable life history strategies, namely mean generation time and death rate. Increasing death rate results in more population turnover each generation and more rapid fixation of GDMI. Panel A shows results for $\mu = 0.25$ while panel B shows results for $\mu = 0.75$. Similarly, shorter generations yields more rapid fixation of GDMI in 10 years because more generations occur within the time window. Panels C and D give show results for a mean generation time of 1.5 months (80 generations in 10 years) in contrast to 3 months (40 generations in 10 years). Panel C maintains $\mu = 0.25$, and panel D maintains $\mu = 0.75$. Fig G. Invasion analyses for variables that influence the probability of invasion. Other parameters are held at their default value according to Table A in S1 Text, while the reproduction number is calculated as selfing rate varies. Lighter areas indicate higher reproduction numbers, and white lines represent the isocline at threshold conditions ($R_0 = 1$). The ratio reported in equation 77 and $R_0$ share a value of 1 under threshold conditions but are otherwise not precisely equal due to the nature of overlapping generations in the model. Fig H. The probability of extinction within 40 generations according to absolute fitness and the number of seeded GDMI individuals. Darker values represent low likelihood of extinction. Fig I. The spread of GDMI in a population with fluctuating carrying capacity due to seasonal rainfall and habitat variation. High seasonality assumes at 4 fold change in carrying capacity in 2 generations, with a full cycle occurring in 4 generations (equal to 1 year with default generation time):

200%, 100%, 50%, 100% carrying capacity cycle. Low seasonality assumes no fluctuation in carrying capacity. Fig J. Simulation of the emergence of a schistosomiasis epidemic under default conditions. GDMI is not present, and long-term behavior of the model is observed to overshoot endemic equilibrium conditions and return to equilibrium over the course of many years. Susceptibility in snails is advantageous at low levels of infection early in the epidemic and is disadvantageous above equilibrium conditions. Fig K. Comparative results among three treatment regimes under high and low transmission conditions. *b* is half of default conditions (left) and $R_0$ = 2.3, producing slower rebounds after annual MDA treatment. More rapid rebounds are observed when *b* is twice default conditions and $R_0$ = 4.5 (right). Fig L. Comparative results among three treatment regimes under high and low intensity MDA application in the human population. 40% annual reduction in MWB (left) produces slower elimination across all treatment regimes compared to 80% annual reduction (right). Rebounds are concave down and relatively smaller for lower intensity MDA and concave up for high intensity MDA. This reflects slower loss of immunity, and for joint treatment the faster gain of GDMI, in the snail population due to higher selection pressure in favor of immunity in higher transmission conditions. Fig M. Simulations of the three treatment regimes for 40 years. Regimes are continued annually for the duration of the simulation (left). MDA is stopped after 10 years of treatment (middle). GDMI is added five years after existing MDA treatment (right).
(PDF)

## Acknowledgments

We thank members of the De Leo lab and ECHO laboratory (Ecole Polytechnique Fédérale de Lausanne) for continued support in this work. We also thank John Pringle and Erin Mordecai for insightful comments on this work.

## Author Contributions

**Conceptualization:** Richard E. Grewelle, Javier Perez-Saez, Josh Tycko, Erica K. O. Namigai, Giulio A. De Leo.

**Data curation:** Richard E. Grewelle.

**Formal analysis:** Richard E. Grewelle.

**Funding acquisition:** Richard E. Grewelle, Giulio A. De Leo.

**Investigation:** Richard E. Grewelle, Javier Perez-Saez, Josh Tycko, Erica K. O. Namigai, Chloe G. Rickards, Giulio A. De Leo.

**Methodology:** Richard E. Grewelle, Javier Perez-Saez, Giulio A. De Leo.

**Software:** Richard E. Grewelle.

**Supervision:** Javier Perez-Saez, Giulio A. De Leo.

**Validation:** Richard E. Grewelle, Chloe G. Rickards.

**Visualization:** Richard E. Grewelle.

**Writing – original draft:** Richard E. Grewelle.

**Writing – review & editing:** Richard E. Grewelle, Javier Perez-Saez, Josh Tycko, Erica K. O. Namigai, Chloe G. Rickards, Giulio A. De Leo.

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
