## [Decision Letter · Decision Letter 0]

6 Sep 2022

Dear Dr. Grewelle,

Thank you very much for submitting your manuscript "Modeling the efficacy of CRISPR gene drive for schistosomiasis control" for consideration at PLOS Neglected Tropical Diseases. As with all papers reviewed by the journal, your manuscript was reviewed by members of the editorial board and by several independent reviewers. The reviewers appreciated the attention to an important topic. Based on the reviews, we are likely to accept this manuscript for publication, providing that you modify the manuscript according to the review recommendations. 

Sincerely,

Gabriel Rinaldi

Academic Editor

Timothy Geary

Section Editor

Reviewer's Responses to Questions

**Key Review Criteria Required for Acceptance?**

**Methods**

-Are the objectives of the study clearly articulated with a clear testable hypothesis stated?

-Is the study design appropriate to address the stated objectives?

-Is the population clearly described and appropriate for the hypothesis being tested?

-Is the sample size sufficient to ensure adequate power to address the hypothesis being tested?

-Were correct statistical analysis used to support conclusions?

-Are there concerns about ethical or regulatory requirements being met?

Reviewer #1: The described methods are appropriate in testing the stated hypotheses and I have no recommendations for further investigation or analyses, however, I have a couple of points for clarification:

The authors state the rationale for not including human immunity in the presented model is that human immunity is unlikely to vary significantly over the 10-year time course of the model, yet several studies have reported immune responses sufficient enough to provide protection in young children (<10years old) in high transmission intensity areas. The literature does not however consistently support a role for human immunity within epidemiological models of Schistosoma infection. Perhaps rephrase the rationale to indicate this.

The authors mention sensitivity analysis and calibrating the model in order to obtain estimates of some parameter values, however I cannot see the details of the method used in the manuscript or supporting information file, only the resulting figure, Fig.4.

Reviewer #2: Please see general comments

**Results**

-Does the analysis presented match the analysis plan?

-Are the results clearly and completely presented?

-Are the figures (Tables, Images) of sufficient quality for clarity?

Reviewer #1: The results and figures are clearly and well presented.

Reviewer #2: Please see general comments

**Conclusions**

-Are the conclusions supported by the data presented?

-Are the limitations of analysis clearly described?

-Do the authors discuss how these data can be helpful to advance our understanding of the topic under study?

-Is public health relevance addressed?

Reviewer #1: The points above are sufficiently addressed, however, it may be useful/interesting for the authors to comment on the species of schistosome that are transmitted by the respective snail species discussed in the manuscript. For example, are those snail species that transmit S.haematobium or S. mansoni more likely to be successful in maintaining gene drive? Are those snails that have low rates of selfing also those that have ecological niches in small water bodies?

Reviewer #2: Please see general comments

**Editorial and Data Presentation Modifications?**

Reviewer #1: Line 68 - '[ref11, 32]', requires edit [11, 32]

Line 123 - 'Tennessen et al. 2015 39', edit reference notation [39]

Line 193 - notation error

Reviewer #2: Please see general comments

**Summary and General Comments**

Reviewer #1: Overview

The authors present a model of snail population dynamics embedded within a standard Schistosoma transmission model. This work forms the basis of a theoretical modelling paper under the untested assumption that GD technology works (i.e. can successfully be introduced into the snail population).

The main findings presented suggest that gene drive mediated immunity is achievable within 10 years of introduction in snail species that demonstrate a low-moderate level of self-fertilisation, out-crossing increases the rate of uptake of gene drive and high rates of self-fertilisation are a contraindication to the establishment of GDMI. The results suggest that GDMI is most likely to be successful in isolated water bodies with small snail populations and contributes to maintenance of reduction combined with MDA is likely to be most successful.

The authors propose potential screening of prevalent snail populations to assess the likelihood of successful implementation of GDMI according to selfing rates in intervention areas.

The paper provides an interesting addition to the schistosomiasis modelling literature, where there is an increasing focus on the importance of snail infection and population dynamics. The manuscript is well-written and the model description is relatively easy to follow. The supporting information provides sufficient descriptions to follow the methodology used, with points for clarification outlined above.

Reviewer #2: The authors present a novel mathematical model to assess the impact of a gene drive in snails which confers immunity to schistosomiasis. A key aspect of this model is bringing together parasite transmission and snail population dynamics, with a population genetic framework which accounts for the capacity of several species of snails to either "self" or reproduce sexually. 

The modelling framework appears technically sound, though this is within the limits of an unpaid review which prohibits exhaustive examination of the 91 supplementary equations, 10 supplementary figures and 800 lines of python code.

Overall the manuscript is well presented and a novel contribution to the field. In my opinion the study is suitable for publication in PLoS NTDs after corrections made for the following points:

1. The title is a little misleading and could be amended to reflect that the proposed genetic engineering is designed to modify a snail's immunity to schistosomiasis (rather than producing sterile snails or some other trait). My understanding is that CRISPR is currently the best, but not the only technique by which this could be achieved. Therefore "Modelling the impact of a gene drive for enhanced snail immunity on schistosomiasis control" would be more accurate.

2. Line 17 - "several forms of cancer", S. haematobium is a causative agent of bladder cancer, I'm not aware of any other types of cancer linked to schistosomiasis.

3. Lines 95-99: it would be helpful to provide at least one or two equations in the main text (even if simplified) which capture the key aspects of snail population dynamics.

4. Equations 5 and 6: In the epidemiological model I'm confused by the "pre-treatment force of infection" term Lambda. The force of infection on snails should be a function of the adult worm burden in humans (w). The eggs excreted into water sources, which perpetuates the lifecycle, is proportional to the adult worm burden in humans. Therefore the force of infection on snails is greater at higher worm burdens and vice versa. I would question therefore the authors' decision to "fix" lambda at a certain value.

5. A "negative binomial distribution of adult worms in the human population" is described in the legend of Figure 1, but this is not elaborated in the methods or the model equations in the main text. The negative binomial parameter k is particularly difficult to estimate from data and the subsequent parasite population dynamics are sensitive to values of k. This should be described in the manuscript. 

6. Line 123 "Biomphalaria glabrata", this is the first mention of a specific species. In this case B. glabrata is the intermediate host for S. mansoni, though prior to this the model equations have only been described generally in terms of "schistosomiasis" rather than a specific species of parasite. Is the subsequent model validation parameterised for S. mansoni? Which species-specific parameters were used? More generally it is unclear whether the model described is suitable for all species of schistosomes or is specific to S. mansoni; this should be clarified in the text.

7. Line 136 "Otherwise, parameters were unaltered from simulation conditions presented throughout the paper"; it's unclear what these parameter values are. I suggest inclusion of a "key parameter table" (not necessarily exhaustive) which contains the most important parameters in the model and the values selected for the simulations with justification (references etc).

8. Figure 2 is important and should be improved. The figure titles would be better as "Snails challenged with X miracidia". The y-axis shows the "Proportion of infected snails". Instead of lines the empirical data could be shown as points with 95% binomial sampling error. This would be more informative than the R2 values given in the legend. The 95% confidence interval from Figure 3B (varying the cost of inbreeding) could also be included in the model estimate in this Figure.

9. Were model parameters fitted to the empirical data, or does Figure 2 represent more of a qualitative comparison? Again it is somewhat unclear how the parameter values were chosen for the model.

10. The "rate" of self-fertilisation is actually a probability.

11. Figure 3A. I would recommend removing the points and adding the 95% confidence interval from the cost of inbreeding. For panel B, choose a colour other than blue or purple for the line (as different to panel A). It's a nice idea to include the actual selfing rates for species of snail, but the clarify of the figure could be improved by spreading the bars apart, choosing a different colour for each species, and breaking up the text. 

12. Figure 4, this is more of a sensitivity analysis than showing "endemic conditions and genetic design". Most of the parameters shown here are not particularly intuitive as they have not previously been introduced. Perhaps a scenario based analysis would be more informative. E.g. the "seed gene drive frequency" is likely to be correlated in real life with the "number of releases"; these could be combined as some measure of "effort" on behalf of the control program. "Gene flow" is dependent on the size of the snail population which is subject to introduction of engineered snails, so again this would be correlated with "effort" in a real world setting.

13. Line 241 "The pre- treatment prevalence of infection is assumed to be 80%"; is this in snails or humans? The observed prevalence of schistosome infections in natural populations of snails is >0.1% even in high transmission settings (e.g. Lake Victoria). What is the presumed worm burden in humans?

14. Figure 5B - the key epidemiological finding of the paper is somewhat hidden at the end. The impact of the gene drive only becomes apparent ~6 years after the combined onset with MDA. This should be mentioned in the abstract. As MDA is ongoing in almost all endemic settings to some extent, another scenario to model would be adding gene drive in the context of MDA which has taken place over the previous 5-10 years. 

15. Why does the mean worm burden increase after 6 years with only MDA, are you assuming reduced uptake or reduced efficacy or praziquantel?

16. The use of "MWB" as an abbreviation for mean worm burden is somewhat unusual. In the legend for Figure 5 this should be written out in full. 

17. Are snails with the naturally immune / engineered genotype fully or partially immune to schistosomes? If fully immune is this biologically plausible?

PLOS authors have the option to publish the peer review history of their article (what does this mean?). If published, this will include your full peer review and any attached files.

Reviewer #1: Yes: Rebecca Claire Oettle

Reviewer #2: Yes: Thomas Crellen

Figure Files:

Data Requirements:

Reproducibility:

References

---

## [Editor Report · Decision Letter 1]

14 Oct 2022

Dear Dr. Grewelle,

We are pleased to inform you that your manuscript 'Modeling the efficacy of CRISPR gene drive for snail immunity on schistosomiasis control' has been provisionally accepted for publication in PLOS Neglected Tropical Diseases.

Best regards,

Gabriel Rinaldi

Academic Editor

Timothy Geary

Section Editor

The authors have successfully addressed all reviewers' comments, and I am happy to accept the manuscript for publication in plos NTDs.

---

## [Editor Report · Acceptance letter]

24 Oct 2022

Dear Dr. Grewelle,

We are delighted to inform you that your manuscript, "Modeling the efficacy of CRISPR gene drive for snail immunity on schistosomiasis control," has been formally accepted for publication in PLOS Neglected Tropical Diseases.

Best regards,

Shaden Kamhawi

co-Editor-in-Chief

Paul Brindley

co-Editor-in-Chief
